# Temperature Dependence on Tensile Mechanical Properties of Sintered Silver Film

**DOI:** 10.3390/ma13184061

**Published:** 2020-09-13

**Authors:** Keisuke Wakamoto, Yo Mochizuki, Takukazu Otsuka, Ken Nakahara, Takahiro Namazu

**Affiliations:** 1ROHM Co., Ltd., Kyoto 615-8585, Japan; takukazu.otsuka@dsn.rohm.co.jp (T.O.); ken.nakahara@dsn.rohm.co.jp (K.N.); 2Faculty of Engineering, Uzumasa, Kyoto University of Advanced Science, Kyoto 615-8577, Japan; namazu.takahiro@kuas.ac.jp; 3Department of Mechanical Engineering, Yakusa, Aichi Institute of Technology, Toyota 470-0392, Japan; yo.mochizuki@rohm.co.jp

**Keywords:** tensile mechanical properties, *S-S* curves, sintered silver, porosity, fracture mechanism, reliability design

## Abstract

This paper investigates the influence of temperature on tensile mechanical properties of sintered silver (s-Ag) films with 8–10 μm in thickness for fundamental reliability design of semiconductor systems. The s-Ag film sintered under a pressure of 60 MPa possesses the porosity (*p*) around 5% determined from cross-sectional scanning electron microscope (SEM) images. The stress–strain (*S-S*) curves of s-Ag and pure silver (p-Ag) films are obtained using originally designed uniaxial tensile tester at temperatures from 25 °C to 150 °C. The S-S curves of p-Ag indicate ductile behavior irrespective of temperature, whereas those of s-Ag indicate brittle-ductile transition at 120 °C. Compared with p-Ag, s-Ag possesses low Young’s modulus (E) and high ultimate tensile strength (UTS) below 80 °C. The mechanism of brittle-ductile transition is discussed based on fracture surface observation results.

## 1. Introduction

Recently, the great demand for high power density of next generation power module packaging has widely increased in power electronics fields. Power modules comprising many parts possess circuit, heat dissipation, and insulation functions. Each part connects by die-attach materials to keep power module function. Power modules have to pass the thermal reliability test, which test temperature fluctuates cyclically. During the test, the coefficient difference between each material generates thermal mechanical stress, causing mechanical damage in the die-attach layers. The shrinkage of die-attach layer makes heat dissipation decreasing, consequently, power modules lead to failure. A die-attach material having good thermal conductivity and mechanical and thermal durability is strongly required.

Solder materials, which are traditional die-attach materials, cannot be applied as is the case for next generation power modules, because the thermal conductivity is technically limited at below around 50 W/m·K. Recently, sintered silver (s-Ag) attracts much attention as a new die-attach material for power electronics owing to its high thermal conductivity [1,2,3,4] and chemical inertness. The mechanical properties as well as thermal and chemical properties play a critical role for the reliability of systems. However, by virtue of including many pores in s-Ag, it is very difficult to finely estimate the mechanical behaviors. To date, many researchers have tried to examine how porous affect the mechanical properties of s-Ag [5,6,7,8,9,10,11,12,13,14,15,16,17,18,19] by means of indentation test [20] and bending test [21,22,23]. Although the tensile test is one of the most basic mechanical property evaluation tests, these alternative mechanical tests have been utilized for the evaluation due to its technical simplicities. We investigated its porosity-dependent tensile mechanical properties of s-Ag thin films (8–10μm thick) as a function of the size of the pores therein [11], and revealed that a porosity of around 5% was good as mechanical structural material. In addition, some researchers have tried to reveal the temperature dependency of the tensile mechanical properties of s-Ag [12,13,14,15,16,17,18,19]. C. Webber et al. performed the tensile tests of s-Ag (*p*: 5%) at room temperature (RT), 125 °C, 200 °C and found that the stress–strain (*S-S*) curves changed from brittle to ductile at 125 °C [14]. C. Chen et al. also investigated s-Ag (*p*: nearly = 25%) temperature dependence on its mechanical behaviors by tensile tests at temperatures from RT to 300 °C [16]. s-Ag (*p*: nearly = 25%) made from micro-sized silver particle showed no clear transition brittle to ductile behavior.

As understood in these previous works, the mechanical behaviors of s-Ag are fully influenced by ambient temperature; therefore, temperature dependence on s-Ag mechanical characteristics must be investigated precisely for reliable design of power modules with s-Ag die-attach material.

In this study, we investigate the influence of test temperature on the mechanical characteristics of s-Ag (*p*: 5%) and p-Ag thin films with 8–10 μm in thickness. The originally designed tensile tester [24,25,26,27] is used to characterize the stress–strain (*S-S*) behaviors of these films at temperatures ranging from RT to 150 °C. The fracture mechanism at each temperature is discussed in the light of fracture surface observation using a scanning electron microscope (SEM).

## 2. Materials and Methods

Figure 1 illustrates a process flow for preparing the tensile test specimen of s-Ag film. A paste with silver particle covered by an organic stabilizer was stencil printed on a metal plate as shown in Figure 1a. The paste includes spherical shape silver particles with the mean size of 20 nm suspended in an organic solvent. Then, the organic solvent was evaporated at 140 °C or 60 min, as depicted in Figure 1b. Then, the dried silver pastes were sintered at 300 °C for 10 min under vertical pressure of 60 MPa via a buffer sheet. Since silver pastes include organic solvents, carbon might affect bonded characteristics after sintering. According to the EDX result on the surface of s-Ag film, carbon is found to be detected in the high porous area greater than 25% porosity where the residual organic stabilizer is still remained, although the data is not shown here. Such the porous state does not enable us to do sintered silver bonding. Under the sintering condition used in this study, the porosity was around 5%, so that carbon was not detected from the s-Ag sheets. Films were prepared as a 20-mm square sheet with 8–10 μm in thickness. A thin oxide layer of around 5 nm in thickness is typically formed on the s-Ag surface [28,29]. However, the oxide layer thickness is approximately 1/200 of the whole thickness of s-Ag film. Therefore, the effect of the oxide film against its mechanical property has not been considered in this study. After the sintering process, as shown in Figure 1c, the s-Ag films were removed from the metal plate, followed by cutting with ultra violet laser to shape a tensile test specimen like a dog-bone, as shown in Figure 1d. The specimen was bonded with a glue to a Si frame with 300 ± 5 μm in thickness to support the dog-bone specimen, as shown in Figure 1e. Precisely alignment between specimen length direction and frame’s tensile direction is important to accurately obtain the mechanical behaviors of the specimen. The misalignment angle within 1.5° was permitted for the tensile test in this study.

The p-Ag specimen was also prepared as the reference material to discuss the influence of pores in the s-Ag specimen on the mechanical characteristics. A 10 μm thick p-Ag sheet having the purity and mean grain size of 99.99% and 0.8 μm, respectively, was made by rolling, followed by cutting with ultraviolet laser, to shape the same configuration to s-Ag specimen. 

The tensile test equipment used here is shown in Figure 2. The tensile tester originally designed for thin films was set into a vacuum chamber. The tester consists the piezoelectric actuator (PI Japan: P-843.40V, Tachikawa, Japan) for applying tensile force, the load cell (Tech Gihan: DGRV10-2N) for tensile force measurement, the micro heater (Hakko Electric Machine Works Co. Ltd.: HLK1151, Nagano, Japan) for temperature control, and the CCD camera for specimen elongation measurement. The test system enables us to control test temperature, tensile strain rate, and test mode (quasi-static, cyclic, stress-relaxation, and creep). The test condition in this study is summarized in Table 1.

A focused ion beam (FIB: FEI, Helious10-G4) was utilized to prepare specimens for cross-sectional observation of s-Ag. The FIB acceleration voltage and beam current were set to be 30 kV and 2400 pA, respectively. The scanning electron microscope (SEM: Hitachi high technology, S4800, Tokyo, Japan) was used for the observation to determine *p*. The electron acceleration voltage (Vea) of 2 kV and a magnification rate (Rmag) of 5000 times were used. The observation angle was 52° against the sample surface to observe the microscopic structure of the s-Ag films clearly. SEM at Vea 1.5 kV was used for fracture surface observation of the samples.

## 3. Results

### 3.1. Determining p of the s-Ag

The s-Ag of *p* was calculated based on cross sectional SEM images because the method could give enough information about microstructure of s-Ag without any troublesome [9,11,14]. Figure 3 shows representative cross-sectional SEM image of s-Ag along with the binary image after filtering. An image area for *p* calculation was set as x = 20 μm and y = 4 μm. To quantify *p*, the observed SEM image was converted to a black-and-white binary image. The black-colored area is identified as the pore area. *p* is calculatedly determined to be as the ratio of the black area to the whole cross-section area. For example, *p* of the presenting sample in Figure 3 is around 5%. 

### 3.2. S-S Curves of the s-Ag and p-Ag

Figure 4a shows representative *S-S* curves of s-Ag at RT, 60 °C, 80 °C, 120 °C, and 150 °C. The *S-S* curve of s-Ag at RT shows almost linear until failure, which indicates that the specimen fractured in a brittle manner during elastic deformation. With temperature elevation, a non-linear region can be found. At 60 °C and 80 °C, the region is seen by strain of a few % just before fracture, which is considered a plastic deformation region. The plastic region drastically increases with increasing temperature to 120 °C and 150 °C. That is, the *S-S* curves of s-Ag indicate that the apparent brittle-ductile transition of s-Ag exists at around 80–120 °C. Figure 4b shows p-Ag representative *S-S* curves for RT, 60 °C, and 120 °C. All the *S-S* curves of p-Ag indicate ductile behaviors in that possess necking region and large plastic region. p-Ag deformed plastically even at temperatures lower than 120 °C.

Figure 5 depicts the obtained mechanical properties ((a): Young’s modulus (E); (b): ultimate tensile strength (UTS); and (c): breaking strain (BS)) as a function of temperature. We defined the E as the slope of stress–strain relation in stress ranging from 30% to 60% of UTS for each specimen. The mean value of s-Ag at RT shows about 45 GPa, which is 27% lower than the p-Ag value. Regarding the temperature dependence on E, the decreasing rate of E with temperature for p-Ag is similar to that for s-Ag. The s-Ag UTS value at RT is 370 MPa, approximately 2.3 times higher than the p-Ag value. The average grain size of p-Ag and s-Ag is 0.8 μm and 0.1 μm, respectively. Therefore, in addition to porosity, grain size becomes one of the material parameters to influence the mechanical characteristics of silver film. That is, based on the Hall-Petch law [30,31], it is reasonable that the UTS of s-Ag was higher than that of p-Ag. The s-Ag UTS value rapidly and linearly decreases with increasing temperature. On the other hand, that of p-Ag slightly decreased with increasing temperature. The phenomenon of the temperature dependence on UTS for p-Ag is quite similar than that for bulk silver, based on the previous study [14].

The value of s-Ag and p-Ag overlaps at around 120 °C. The temperature dependency of BS for s-Ag shows a trend similar to that for p-Ag in that the BS value shows positive correlation with increasing temperature. Compared with p-Ag, the BS value of s-Ag rises drastically when increasing temperature from 80 °C to 120 °C. 

## 4. Fracture Mechanism

Figure 6a–c show representative SEM images of fracture surface for p-Ag, along with the magnified images after the tensile tests at RT, 60 °C, and 120 °C, respectively. The fracture surface at RT possesses a big protrusion at the middle of the specimen, which indicates that the specimen showed ductile fracture after necking during tensile loading. Irrespective of temperature elevation, the entire shape of the fracture surfaces unchanged, although the number of small concave points on fracture surface increases with an increase of temperature. Therefore, all the p-Ag specimens fractured plastically within the temperature range tested. On the other hand, s-Ag specimens showed a different trend in fracture with p-Ag specimens, as shown in Figure 7a–c indicating the representative fracture surfaces after the tests at RT, 60 °C, and 120 °C, respectively. At RT, the fracture surface shows irregularity compared with that for p-Ag specimen. Many concaves can be seen on the fractured surface, which are dimples that appeared by locally occurring ductile deformation. Even with increasing temperature to 60 °C, no remarkable change is found on the fractured surface. Note that the fractured surface at 120 °C, completely differs from those at RT and 60 °C. In the front view of the fractured surface, a convex-concave pattern like a family of grains can be found. However, in the magnified image of the side view, the fractured surface is found to consist of a lot of sharpened protrusions facing to the tensile direction. The difference in fracture mechanism between s-Ag and p-Ag would be related to microstructure. The s-Ag specimen contains many silver nanoparticles, having the higher surface area-to-volume ratio. An Ag nanoparticle is known to provide high surface energy [32], leading to high binding energy between s-Ag grains under the appropriate sintering process condition. In addition, smaller grain size of s-Ag also affects the tensile strength in accordance with Hall-petch strengthening phenomena [30,31]. At low temperature, therefore, only very local plastic deformation inside grains would have happened because of strengthened grain boundary, and a small dimple pattern for each grain would have been produced on the whole fractured surface; consequently, strengthened specimens showing brittle-like fracture would have been produced. At high temperature, since the flow ability of each Ag grain increased, large plastic strain would have been obtained as with the p-Ag specimens. However, the brittle-ductile transition mechanism should not be concluded without dislocation observation by using TEM observation.

## 5. Conclusions

We investigated the influence of test temperature on tensile mechanical properties of s-Ag thin films with 8–10 μm in thickness. All the films sintered at 60 MPa possessed *p* around 5%. Quasi-static tensile tests were performed using an in-house tensile tester at temperatures from RT to 150 °C. The *S-S* curves of p-Ag indicated ductile behavior irrespective of temperature, whereas those of s-Ag indicated brittle-ductile transition at around 120 °C. The E of s-Ag was lower than that of p-Ag at below 100 °C, whereas the UTS of s-Ag was higher than that of p-Ag at the same temperature range. These facts suggested that s-Ag was more deformable elastically and stronger than p-Ag, the characteristics of which would be useful for reliable and advanced design of future power semiconductor packages.

## Figures and Tables

**Figure 1 materials-13-04061-f001:**
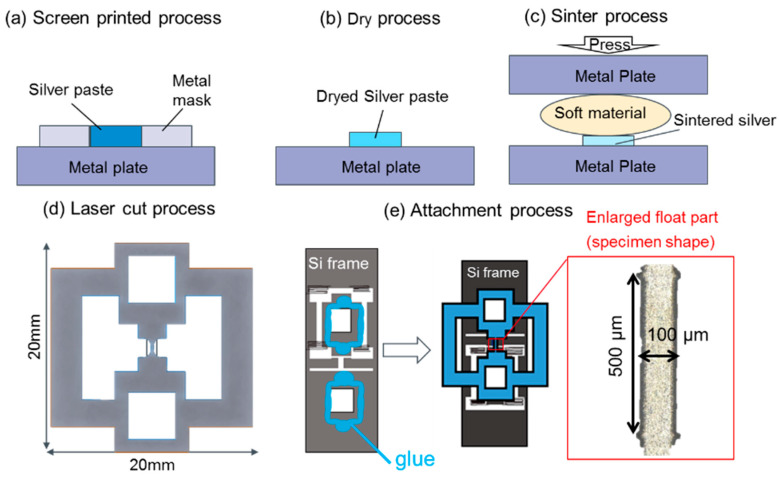
The sample preparation process used here: (**a**) screen printing, (**b**) dry process, (**c**) sintering with uniaxial press, (**d**,**e**) show the sample shape: (**d**) silver film after laser cut, and (**e**) the attachment process.

**Figure 2 materials-13-04061-f002:**
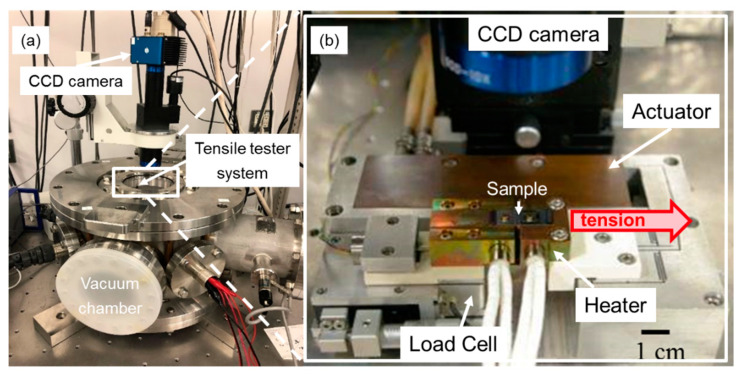
Tensile tester system (**a**): overall equipment image (**b**): magnified tensile tester system.

**Figure 3 materials-13-04061-f003:**
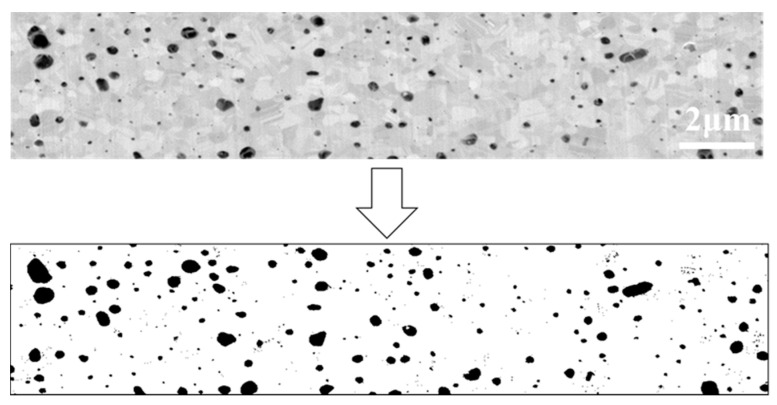
Cross-section scanning electron microscope (SEM) and the filtering images of s-Ag (upper side: cross section SEM image, bottom side: binary image after filtering the SEM image).

**Figure 4 materials-13-04061-f004:**
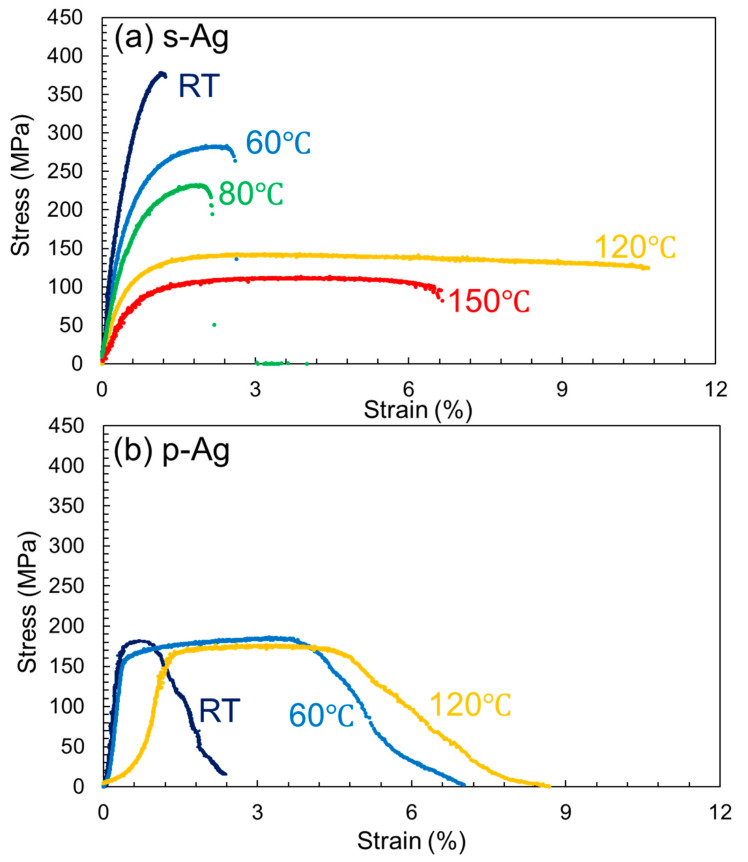
Stress–strain (*S-S*) curves for each temperature (**a**) s-Ag (**b**) p-Ag.

**Figure 5 materials-13-04061-f005:**
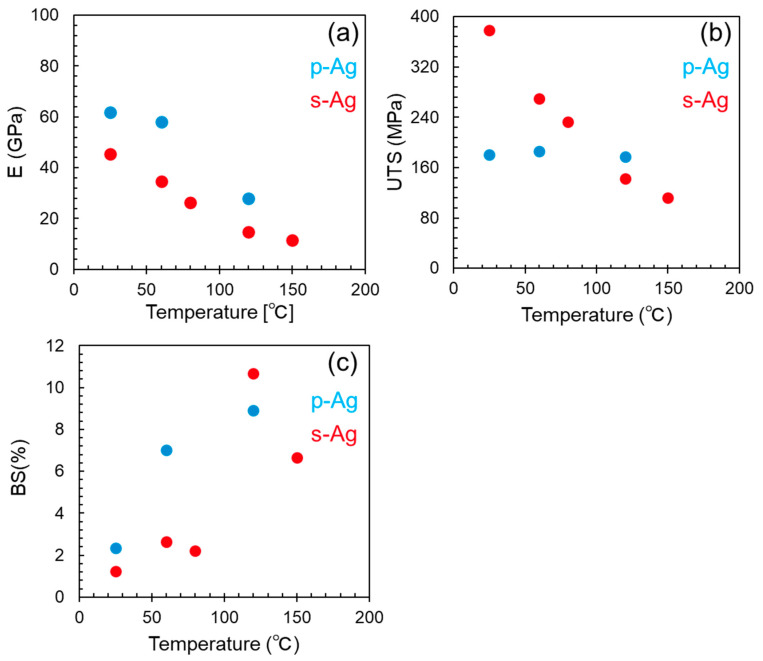
(**a**) Young’s modulus €, (**b**) ultimate tensile strength (UTS), and (**c**) breaking strain (BS) as a function of temperature. The red and blue-colored points denote the data of s-Ag and p-Ag, respectively.

**Figure 6 materials-13-04061-f006:**
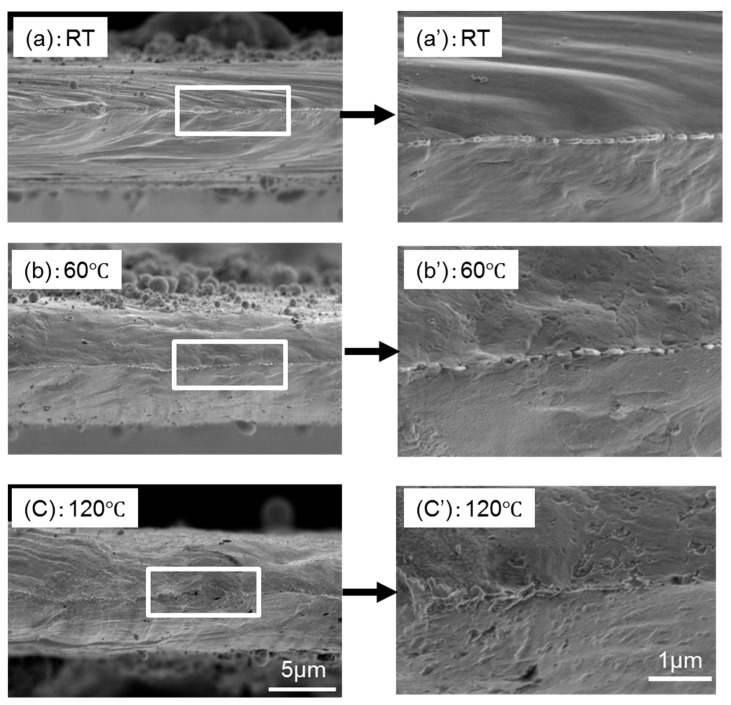
(**a**–**c**) Representative SEM images of p-Ag fracture surface together after tensile tests at room temperature (RT), 60 °C, and 120 °C, respectively. (**a’**–**c’**) Enlarged SEM images of each area denoted by white square shown in Figure 6a–c.

**Figure 7 materials-13-04061-f007:**
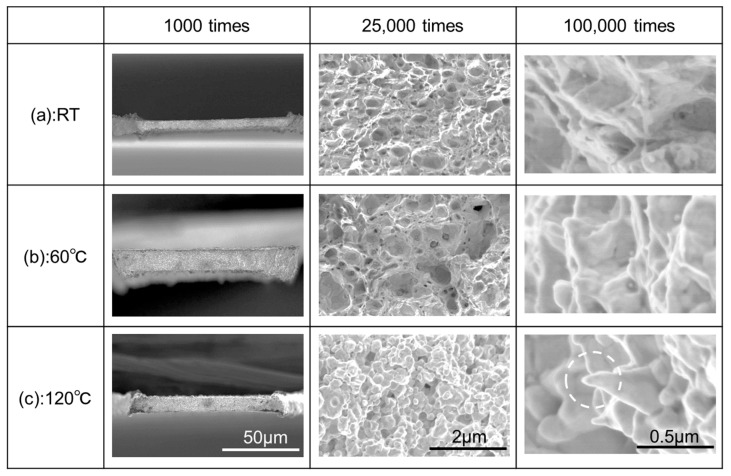
(**a**–**c**) Representative SEM images of s-Ag fracture surface together with the magnified images after tensile tests at RT, 60 °C, and 120 °C, respectively.

**Table 1 materials-13-04061-t001:** Tensile test conditions.

Material	Test Temperature (°C)	Strain Speed (s^−1^)
s-Ag	RT, 60, 80, 120, 150	1 × 10^−4^
p-Ag	RT, 60, 120	

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
