# Peer review of "Temperature Dependence on Tensile Mechanical Properties of Sintered Silver Film"

_materials, 2020, doi:10.3390/ma13184061_

Round 1

Reviewer 1 Report

The subject of the paper is of practical interest. The experimental work is well design and conducted. Investigation results are supporting the conclusions.

I suggest elaborating in section 2 Materials and Methods the following part: “The p-Ag film specimen  was also prepared as the reference material”, since there are little information about it.

Reviewer 2 Report

This manuscript investigated the influence of temperature in the range of 25 to 150°C on the tensile mechanical properties of sintered silver thin films. The brittle-ductile transition at 120°C was identified from the stress-strain curves. The manuscript is well-written and well organized. Albeit the content of the manuscript is relatively concise, it clearly presented its experimental results and discussed the underlying mechanism of this phenomenon, which should have met the publishing requirements. I would recommend publishing this manuscript after addressing the following comments:

  1. What is the average size of the silver nanoparticles used to prepare the sintered film? This is important parameter since finer grains should have a higher impact toughness and lower ductile-brittle transition temperature.
  2. For sintered materials, in addition to porosity, grain size and other parameters should also have an important impact on the ductile-brittle transition temperature, and relevant references should be given in the introduction.
  3. Please give the full name of room temperature (RT) when it first appears.
  4. Line 60, "Fig. 1" should be "Figure 1" because it appears at the beginning of the sentence.

Reviewer 3 Report

Q1. Please describe the detail information for pure Silver

Q2. It seems necessary to explain how the pore density was calculated or to add a reference.

Q3. As the author mentioned in the main text, grain size is also considered to be an important factor determining the mechanical properties of the metal. Could you compare the grain size of p-Ag and s-Ag?

Q4. Comparing the SEM images in Fig. 6 and Fig. 7 with each other at the same magnification will help to understand the mechanism.

Q5. Lastly, have the author tried chemical composition analysis for s-Ag?

The author use silver-paste as the source for s-Ag. Even if all of the solvents ㅈwere vaporized, the organic matter will still remain and there is a high probability that it will remain in the sintering process. In particular, at about 300 oC, carbon could be diffuse into the Ag or carbonized on the surface.

Reviewer 4 Report

Comments and Suggestions:

The mechanical testing setup coupled with temperature control is great!

Major issues:

  1. What is APPARENT Young’s modulus?
  2. The presented Young’s modulus of pure silver film in Fig. 5a does not match slope for the linear region of stress-strain curves of pure silver film in Fig. 4b. The slopes for linear regions are similar between room temperature, 60 ºC, and 120 ºC, while the Young’s modulus decreases significantly as temperature increases.
  3. Here are a few questions and suggestions for the fracture mechanism:
  • SEM images are not enough to support analysis of fracture mechanism. The sintered silver under pressure could have more dislocations, while heating treatment would reduce dislocation density. The dislocation could account for the brittle-ductile transition.
  • Regarding that the film thickness is less 10 micrometers, if there is a thin oxide layer forming during sintering, the mechanical behavior of silver film could be heavily affected. Such possibility has not been ruled out. Associated analysis should be included
  • The reviewer does not get why the pure silver film has a constant ultimate strength with respect to temperature change.
  1. The fracture surfaces are similar between sintered silver film tested under various temperature. Generally, the spike shape could indicate that the material experiences ductile fracture. From the last two SEM images, such spike is rare. More evidences are expected to support the conclusion.

Round 2

Reviewer 3 Report

The authors addressed all my concerns. 

I believe that the manuscript is appropriate for publication.

Author Response

Thank you so much for your comments. I would really appreciate you to improve my paper.  

Reviewer 4 Report

Minor Issues:

  1. The authors changed "Apparent modulus" into "Modulus". Again, several "apparent modulus" can still be found in the manuscript;
  2. From the given three sets data, it is almost impossible to identify the linear relationship between Elastic Modulus and Temperature for p-Ag. As the data show that the E is similar when the temperature is less than 60 degree Celsius. Therefore, it would be better to remove the solid line in Fig. 5a in case of misleading the authors.
